# Effects of Dietary Supplementation of Humic Acid Sodium and Zinc Oxide on Growth Performance, Immune Status and Antioxidant Capacity of Weaned Piglets

**DOI:** 10.3390/ani10112104

**Published:** 2020-11-13

**Authors:** Qi Wang, Jiafu Ying, Peng Zou, Yuanhao Zhou, Baikui Wang, Dongyou Yu, Weifen Li, Xiaoli Zhan

**Affiliations:** 1Institute of Animal Nutrition and Feed Sciences, College of Animal Sciences, Zhejiang University, Hangzhou 310058, China; 13430230952@163.com (Q.W.); 21817401@zju.edu.cn (J.Y.); 21917085@zju.edu.cn (P.Z.); zyh17767072477@163.com (Y.Z.); wangbaikui@zju.edu.cn (B.W.); 2Key Laboratory of Animal Feed and Nutrition of Zhejiang Province, Institute of Animal Nutrition and Feed Sciences, College of Animal Sciences, Zhejiang University, Hangzhou 310058, China; dyyu@zju.edu.cn; 3Key Laboratory of Molecular Animal Nutrition of the Ministry of Education, Institute of Animal Nutrition and Feed Sciences, College of Animal Sciences, Zhejiang University, Hangzhou 310058, China; wfli@zju.edu.cn

**Keywords:** sodium humate, zinc oxide, antioxidant capacity, growth performance, immunity, piglets

## Abstract

**Simple Summary:**

Weaning of piglets can destroy the piglet’s intestinal health and immune function, leading to diarrhea and a reduction in growth rate eventually. Considering drug resistance and residues, new alternatives, such as sodium humate (HNa), have attracted considerable research interest over recent decades. Our study was designed to explore the effect of HNa on the growth performance, diarrhea rate, antioxidative, inflammation, and immunity of weaned piglets and the possibility of HNa replacing antibiotics and zinc oxide. The obtained results indicate that HNa reduces stress, protects the intestinal barrier, and improves the performance of weaned piglets.

**Abstract:**

At present, the widespread use of high-dose zinc oxide and antibiotics to prevent post-weaning diarrhea (PWD) in piglets has caused serious environmental problems. To solve this problem, we studied the effect of HNa as a substitute for zinc oxide (ZnO) and antibiotics on the growth performance, immune status, and antioxidant capacity of piglets. Seventy-two weaned piglets (body weight = 7.42 ± 0.85 kg, 26-d-old) were distributed in a randomized 2 × 3 factorial design (two sexes and three treatments) with six replicates of four piglets each. The three treatments were the control diet (basic diet), HNa diet (basic diet + 2000 mg/kg sodium humate), and ZoA group (basic diet + 1600 mg/kg zinc oxide + 1000 mg/kg oxytetracycline calcium). ANOVA and Chi-square tests were applied to compare the means (*p* < 0.05) between treatments. The results showed that body weight at 16 and 30 d and the average daily gain of piglets fed with HNa or ZoA were significantly higher (*p* < 0.05) than the control group. Supplementing HNa or ZoA significantly increased (*p* < 0.05) the level of immunoglobulin M and G, and reduced (*p* < 0.05) the concentration of inflammatory factors such as tumor necrosis factor-alpha (TNF-α), interleukins IL-6 and IL-1β, myeloperoxidase (MPO), and diamine oxidase (DAO). Furthermore, dietary HNa or ZnO significantly reduced (*p* < 0.05) the level of total antioxidant capacity (T-AOC) and malondialdehyde (MDA) compared with the control group. ZoA treatment showed an upward trend of IgA level and a downward trend of the concentration of lipopolysaccharide (LPS) and catalase (CAT). Overall, the study demonstrated that the addition of HNa in the diet partially replaced antibiotics and ZnO to improve the growth performance, immune function, and antioxidant capacity of weaned piglets, and maintained a good preventive effect on piglet diarrhea.

## 1. Introduction

Weaning of piglets can cause oxidative damage [1], thereby destroying the piglet’s intestinal health and affecting its immune function, eventually leading to diarrhea and reducing growth rate [2,3]. In the past few decades, antibiotics and zinc oxide have been widely applied in the pig industry to reduce piglet diarrhea and improve immunity [4,5,6]. However, the abuse of antibiotics and zinc oxide has also caused problems such as antibiotic resistance and heavy metal residues, and the occurrence of these problems has forced research institutions and breeding personnel to find dependable feed additives.

Humus substances (HS) are a type of compound produced by the decomposition of organic matter in the soil and are often used to solve piglet diarrhea, indigestion, and acute poisoning [7]. HS mainly contain humus, fulvic and ulmic acids, and trace minerals. Sodium humate is the sodium salt of humic acid. It is a product obtained by treating lignite, grass coal, weathered coal, etc., with nitric acid and sodium hydroxide through oxidation, alkali extraction, and acid purification [8]. The molecule contains various mineral elements such as Ca, Fe, Na, Mn, Zn, etc. It has a strong ion exchange ability for heavy metal ions, and also, has strong adsorption, complexation, and chelating ability. At the same time, sodium humate has the advantages of wide sources, low price, and no drug residues, so they have a wide range of applications [9]. Previous studies have shown that HNa as a feed additive in piglets’ diet can promote the secretion of gastric juice in animals and increase appetite, stimulate the growth of beneficial bacteria in the gastrointestinal tract of animals, inhibit the reproduction of spoilage bacteria, and reduce ammonia emissions [1,7,9].

Due to concerns about drug resistance and residues, new alternatives including HNa have attracted considerable research interest in the past decade. However, there are few studies on the effects of HNa supplementation on piglets. Our study was designed to explore the effects of HNa on the growth performance, diarrhea rate, anti-oxidative, inflammation, and immunity of weaned piglets from 26 to 56 d of age and the possibility of HNa replacing antibiotics and zinc oxide. This research will promote the application of HNa as a substitute for traditional antibacterial drugs.

## 2. Materials and Methods

The Institutional Animal Care and Use Committee of Zhejiang University (Hangzhou, China) has approved all procedures for the experiment, with the permit number for conducting animal experiments of ZJU2019-145-85.

### 2.1. Animals, Treatments, and Housing

A total of 72 healthy piglets (Yorkshire × Landrace × Duroc, half male and half female) weaned at 26 days with an initial average body weight (7.42 ± 0.85 kg) were randomly assigned to 3 treatments with 4 replications of 6 piglets in each pen. The specific grouping adopts a completely randomized block design, with gender and weight at weaning as the grouping criteria. Piglets in the 3 dietary treatments were fed a control diet (basal diet without any treatment), ZoA diet (basal diet + 2000 mg/kg Zinc oxide + 1000 mg/kg oxytetracycline calcium), or HNa diet (basal diet + 2000 mg/kg sodium humate). The entire test period lasted for 30 d, and was divided into 2 periods, day 0 to 16 and day 16 to 30. The experimental diets were formulated to meet the requirements for nursery pigs suggested by the United States National Research Council (NRC, 2012) in 2 phases: phase 1 (d1–16) and phase 2 (d17–30). The composition and chemical composition of the basal diets are shown in Table 1.

The composition and nutritional level of the basal diet are listed in Table 1. Piglets were raised in the same environment and were allowed to have free access to feed and water. The refusals were removed before the subsequent feeding, weighed, and taken into account in the calculations of feed consumption. Dietary treatment was maintained for 30 days. The average room temperature and relative humidity were, respectively, maintained at 24 to 26 ℃ and 60% to 70%.

### 2.2. Growth Performance

During the study, the live weights of piglets were determined weekly from 26 to 56 days using a scale with precision ±1 g (Metter Toledo, digital Scale, Shanghai, China). The individual weights of pigs were recorded at day 0, 16, and 30 during the experiment and weighed after fasting. After the end of the experiment, the feed intake of each weaned piglet was recorded every 3 days. On the basis of these values, we calculated the average daily feed intake (ADFI), average daily gain (ADG), and the ratio of feed to weight gain (F:G ratio).

### 2.3. Diarrhea Evaluation.

Piglets were clinically checked once a day. The severity of diarrhea was evaluated by scoring the stool consistency via three independent trained persons: 0—normal; 1—paste; 2—paste; 3—fluids; 4—fluids plus blood. The incidence of diarrhea was assessed by the proportion (%) of scoured piglets in the group. The average daily diarrhea score (DDS) is calculated as the group sum divided by the number of piglets in the group. The duration of diarrhea was recorded individually and the mean duration for the group was calculated. Diarrhea rate was calculated as follows: diarrhea rate (%) = (number of diarrhea piglets × diarrhea days)/(number of total pigs × experiment days) × 100%.

### 2.4. Sample Preparation and Analysis

On the day 0 and day 30 of the experiment, 2 pigs (one male and one female) from each group were randomly selected, and venous blood was obtained through anterior vena cava puncture using 5 mL of ethylenediaminetetraacetic acid dipotassium (EDTAK2). The sample was centrifuged at 3000× *g* for 30 min and stored at −20 ℃ until analysis.

Serum immunoglobulins (IgG, IgM, IgA) and inflammatory factors (LPS, IL-1β, IL-6, IL-10) were determined using commercial enzyme linked immunosorbent assay (ELISA) kits (Elabscience, Wuhan, China). Total antioxidant capacity (T-AOC), diamine oxidase (DAO), myeloperoxidase (MPO), catalase (CAT), endotoxin (lipopolysaccharide, LPS), and malondialdehyde (MDA) active ties were measured via colorimetric methods using a Microplate Reader (SpectraMax M5, Molecular Devices, CA, USA). Assay kits for the tests were purchased from Nanjing Jiancheng Bioengineering Institute (China).

### 2.5. Statistical Analysis

The data related to growth performance, immune status, antioxidant capacity, and diarrhea rate were analyzed by one-way ANOVA with SPSS 20.0 software (SPSS Inc. Chicago, IL, USA). Results were presented as mean ± standard error of the mean (SEM). Moreover, the Chi-square test was used to analyze the incidence of diarrhea. *p* values < 0.05 were considered statistically significant, and 0.05 < *p* < 1 was considered as tending to significance.

## 3. Results

### 3.1. Growth Performance and Diarrhea Incidence

As shown in Table 2, compared with the control group, pigs fed with the HNa or ZoA diet had greater BW (16d), final BW, ADG (days 1–16, days 16–30, and days 1–30), and ADFI (days 1–16) and lower F:G ratio (days 1–16 and days 1–30) and diarrhea rate (days 1–16, days 16–30, and days 1–30, *p* < 0.05). In addition, compared with the HNa group, the ZoA group had better effects on preventing and treating diarrhea in weaned piglets.

### 3.2. Serum Cytokines Concentrations and Immunoglobulin Concentrations

As shown in Table 3, compared with the control group, dietary HNa and ZoA diet significantly decrease concentrations of TNF-α, IL-1β, IL-6, DAO, MPO, and DAO level (*p* < 0.05), increase the concentrations of IgG and IgM. Meanwhile, the ZoA group has significant increases in the level of IgA compared with the control group (*p* < 0.05). However, no significant difference was observed in the concentrations of serum immune index among HNa and ZoA.

### 3.3. Antioxidant Properties

As shown in Table 4, compared with the control group, dietary HNa or ZoA significantly decrease the concentration of MDA (*p* < 0.05). Furthermore, dietary HNa increased the level of CAT and T-AOC compared with other groups.

## 4. Discussion

### 4.1. Growth Performance and Diarrhea Rate

Piglets usually suffer an extensive amount of stress responses after weaning, such as impaired bowel function, and decreased immunity and antioxidant capacity, which in turn leads to reduced feed utilization, growth delay, and increased morbidity and mortality [10]. Zinc is one of the indispensable micronutrients in mammalian diets at lower concentrations (100 to 150 ppm), serving as a cofactor for hundreds of cellular enzymes. The addition of zinc sulfate (100 mg) to the basal diet for weaning piglets is to ensure the healthy growth of weaned piglets and to supplement enough trace elements [4,5,6]. A lot of studies confirmed that dietary ZnO supplementation improves growth rate and decreases diarrhea rate in post-weaning piglets [11,12]. High doses of ZnO (2500 to 3000 mg/kg of feed) have been widely used in post-weaning diarrhea (PWD) prophylaxis and treatment [6,13]. The mechanism of action of zinc is still not fully understood. Although ZnO and antibiotics have traditionally been used in diets for weaned piglets, their excretion in high amounts represents a hazard to the environment [6,14]. To reduce environmental pollution and drug resistance, reducing the dosage of zinc oxide and antibiotics and finding alternatives have become an urgent problem. HNa and ZnO were recommended for the treatment of diarrhea, dyspepsia, and acute intoxications in swine, horses, and poultry. It is reported that HNa can improve the performance of weaned piglets, which may be due to the stability of the intestinal microbiota and the subsequent improvement in nutrient absorption, especially protein digestion and utilization of trace elements [15]. In the past few decades, HNa as an alternative to antibiotics has been used to improve growth performance and reduce diarrhea in piglets [9,16,17]. Kaevska et al. [7] reported that supplementing HNa (2000 mg/kg) in feed can improve ADG and decrease the F:G ratio and diarrhea in weaned piglets.

Similarly, the results of our study have shown that dietary HNa significantly increased ADG and decreased the F:G ratio and diarrhea rate of weaned piglets (*p* < 0.05). This may be due to HNa potentially improving the piglet’s intestinal health, increasing the digestibility of nutrients, and maintaining intestinal microbiota balance and the integrity of intestinal barrier. HNa exerts a protective effect on the intestinal mucosa and has anti-inflammatory, adsorption, antibacterial, and antitoxic effects [18]. However, HNa benefits animal performance and prevents diarrhea, even though the actual mechanism is still unclear [19,20,21].

### 4.2. Serum Cytokines Concentrations and Immunoglobulin Concentrations

The concentrations of IgA, IgG, and IgM are the main indicators reflecting the ability of the animal body to resist various infections. Previous studies have confirmed that HNa and ZnO can improve the immune system of livestock and poultry [9,10]. In the current study, HNa and ZoA significantly increased the concentration of IgG and IgM compared with the control group (*p* < 0.05). However, dietary HNa has no effect on IgA concentration compared with the control group. One possible reason is that under normal feeding conditions, adding HNa to the diet cannot stimulate the body to produce strong mucosal immunity. Furthermore, compared with the control group, the ZoA group had a significant increase in the level of serum IgA. This shows that ZoA in the diet can improve the immunity of weaned piglets by increasing the content of immunoglobulin in serum. However, the specific mechanisms by which HNa and ZoA increase the secretion of immunoglobulin are still unclear, and further research is needed.

In addition, in experimental animal models and humans, cytokine concentration is also widely used to evaluate the response to pathogens and antigens, because it affects immune regulation and disease resistance. A large amount of evidence shows that the expression of pro-inflammatory cytokine genes is upregulated in a short time after weaning. Studies have shown that adding HNa and ZnO to feed can enhance the immune response of early-weaned pigs by regulating the production of inflammatory cytokines (IL-1b, IL-6, TNF-α, and IL-10) [9,22]. Our study also reported that dietary supplementation of ZoA and HNa reduced serum TNF-a, IL-1β, IL-6, DAO, and MPO concentrations. This shows that HNa and ZoA can regulate the immune status of weaned piglets by reducing the production of pro-inflammatory cytokines. This may be because HNa contain a large amount of active substances. After entering the digestive tract, the absorbed components may initiate changes in blood immunological parameters, while the unabsorbed components may help relieve the intestinal immune defense stress. These results indicate that dietary HNa may improve intestinal integrity in part by reducing pro-inflammatory stimuli rather than enhancing anti-inflammatory responses. In addition, LPS produced by the intestinal flora is a key factor in inducing the body’s inflammatory response. It mainly acts as a mediator and can cause the release of downstream inflammatory factors such as IL-6 and TNF-α. Our research results show that zinc oxide and antibiotics can significantly reduce the level of LPS in the serum, indicating that it mainly achieves the purpose of improving immunity and reducing inflammation by inhibiting the abundance of harmful intestinal microbiota.

### 4.3. Anti-Oxidative Effects

Weaning is the most difficult period in a piglet’s life. It is related to social stress and dietary changes [6], and weaning stress can inhibit the activity of antioxidant enzymes and promote the production of free radicals, which leads to the oxidative damage of lipids, proteins, and DNA [1,23]. This oxidative imbalance can lead to immunosuppression with increased sensitivity to various diseases and in negative alteration in the growth performance of animals. Furthermore, oxidative stress caused by weaning may have a negative influence on meat quality and shelf life [21,24]. Zn is an essential trace element important for growth, immunity, and healing, and it has also been shown to have antioxidant properties. Wang et al. [25] showed that high dietary Zn (3000 mg/kg as ZnO for 15 days) reduced levels of oxidative stress and prevented apoptosis in the jejunum of weaned piglets. Similarly, dietary 1000–2000 mg/kg Zn for 14 days was found to affect the expression of genes involved in reducing oxidative stress in newly weaning piglets [26]. However, Gazaryan et al. pointed out that excessive Zn^2+^ can interfere with mitochondrial antioxidant production and may stimulate the production of reactive oxygen species [27]. Our experimental results also show that adding 2000 mg/kg of zinc oxide to the feed can improve the antioxidant capacity of piglets without side effects.

Previous studies have shown that the reason for the mechanism of HNa’s antioxidant activity is the vitamins, antibiotics, alkaloids, and physiologically active substances in it being able to combine with hydrogen peroxide free radicals to reduce or eliminate its activity [28,29]. In this study, compared with the control group, adding 2000 mg/kg of compound HNa to the diet can significantly increase the total superoxide dismutase activity and total antioxidant capacity in the serum of piglets, and reduce the content of malondialdehyde. Weber et al. suggested that HNa in the diet of young pigs (2500 mg/kg for 35 days) may play an important role in eliminating the effects of oxidative stress in the body [30]. Moreover, adding HS to the diets of the diets of broilers and growing pigs was found to reduce lipid peroxidation in muscle tissues during storage, which had a positive effect on meat quality [21,24]. The in vitro study of Vaskova et.al. [31] reported that the superoxide dismutase activity of piglet serum decreased, but the activities of other antioxidant enzymes were not affected, and no HS-supported ROS generation was observed after HNa treatment. However, HNa is a substance with a very complex structure, and the mechanism of its antioxidant capacity is currently unclear, and further research is needed.

## 5. Conclusions

In summary, our results show that dietary supplementation of HNa improves the growth performance, immune function and antioxidant status of weaned piglets and decrease the diarrhea rate. The curative effect of the sodium humate treatment group is very similar to that of the zinc oxide group, indicating that sodium humate can replace or partially replace the application of zinc oxide in the pig industry. However, the specific mechanism of HNa is still unclear, and further research is needed.

## Figures and Tables

**Table 1 animals-10-02104-t001:** Composition and nutrient levels of basal diets (air-dry basis).

Ingredients	(% as Fed)	
	0–16 d	16–30 d
Corn	51	64
Extruded soybean	13	10
Soybean meal	18	20.4
Whey powder	10	0
Fish meal	5	2
CaHPO4	0.8	1.2
Limestone	0.5	0.55
Premix ^1^	1.7	1.85
Chemical composition (%DM)		
Moisture	11.5	11.42
Crude protein	22.33	22.11
Ether Extract	4.82	4.90
Crude Fiber	3.15	2.97
Ash	6.10	7.01
Ca	0.76	0.75
Total P	0.71	0.70
Lysine	1.42	1.44
Methionine	0.51	0.50
Metabolic Energy (Mcal/kg)	3.82	3.78

^1^ The premix provided the following per kg of diets: VA 7 500 IU; VD_3_ 750 IU; VE 25 IU; VK_3_ 2.0 mg; VB_1_ 1.875 mg; VB_2_ 3.75 mg; VB_6_ 2.19 mg; VB_12_ 0.025 mg; nicotinic acid 25 mg; pantothenic acid 15.6 mg; folic acid 2.0 mg; biotin 0.1875 mg; Cu (as copper sulfate) 25 mg; Zn (as zinc sulfate) 100 mg; Fe (as ferrous sulfate) 100 mg; Mn (as manganese sulfate) 30 mg; Se (as sodium selenite) 0.3 mg; I (as potassium iodide) 0.4 mg. while the others were measured values.

**Table 2 animals-10-02104-t002:** Effects of HNa on growth performance of piglets.

Item	Group				
	Control	HNa	ZoA	SEM	*p*-Value
**body weight/kg**					
**0d**	7.42	7.5	7.41	0.431	0.97
**16d**	9.94 ^b^	11.12 ^a^	11.16 ^a^	0.556	0.095
**30d**	13.02 ^b^	15.19 ^a^	14.99 ^a^	0.592	0.009
**ADG/g**					
**1–16**	157.78 ^b^	233.84 ^a^	226.2 ^a^	19.854	0.007
**16–30**	219.42 ^b^	273.64 ^a^	289.79 ^a^	23.064	0.033
**1–30**	186.54 ^b^	252.42 ^a^	255.88 ^a^	12.414	0.001
**ADFI/g**					
**1–16**	288.02 ^b^	321.8 ^a^	328.39 ^a^	25.922	0.296
**16–30**	480.09 ^b^	501.6 ^ab^	585.13 ^a^	45.200	0.1
**1–30**	371.61 ^b^	431.8 ^a^	406.03 ^ab^	31.134	0.208
**F:G**					
**1–16**	1.87^a^	1.43 ^b^	1.41 ^b^	0.137	0.014
**16–30**	2.21^a^	2.03 ^ab^	1.87 ^b^	0.264	0.46
**1–30**	1.98^a^	1.69 ^b^	1.60 ^a^	0.084	0.004
**Diarrhea rate**					
**1–16**	14.90% ^a^	8.20% ^b^	3.00% ^c^	0.567%	<0.001
**16–30**	28.50% ^a^	10.90% ^b^	6.90% ^c^	1.236%	<0.001
**1–30**	21.70% ^a^	9.55% ^b^	4.95% ^c^	0.583%	<0.001

HNa—Sodium humate; ZoA—zinc oxide + antibiotic; ADG—average daily gain; ADFI—average daily feed intake; F:G—feed to gain ratio. ^a, b, c:^ Within a row, values with different letter superscripts mean significant difference (*p* < 0.05).

**Table 3 animals-10-02104-t003:** Effects of HNa on serum cytokines and Immunoglobulin concentrations.

Item	Group				
	Control	HNa	ZoA	SEM	*p*-Value
**LPS (EU/mL)**	0.47 ^a^	0.40 ^ab^	0.37 ^b^	0.031	0.051
**IL-1β** **(pg/mL)**	30.64 ^a^	23.11 ^b^	24.08 ^ab^	1.353	0.001
**IL-6 (pg/mL)**	142.93 ^a^	115.97 ^b^	117.39 ^b^	6.019	0.002
**TNF-α (pg/mL)**	61.82 ^a^	46.72 ^b^	44.41 ^b^	1.715	<0.001
**DAO (U/L)**	25.59 ^a^	18.84 ^b^	20.60 ^b^	1.447	0.003
**MPO (U/L)**	51.41 ^a^	40.11 ^b^	40.47 ^b^	3.450	0.016
**IgG (g/L)**	19.90 ^b^	21.31 ^a^	20.92 ^a^	0.347	0.008
**IgM (g/L)**	2.38 ^b^	2.46 ^a^	2.48 ^a^	0.022	0.030
**IgA (g/L)**	1.10 ^b^	1.26 ^ab^	1.33 ^a^	0.073	0.030

HNa—sodium humate; ZoA—zinc oxide + antibiotic; LPS—lipopolysaccharide; IL-1β—interleukin-1β; IL-6—interleukin-6; TNF-α—tumor necrosis factor-α; DAO—diamine oxidase; MPO—myeloperoxidase; IgG—immunoglobulin G; IgM—immunoglobulin M; IgA—immunoglobulin A. ^a, b,^ within a row, values with different letter superscripts mean significant difference (*p* < 0.05).

**Table 4 animals-10-02104-t004:** Effects of HNa on serum antioxidant function in piglets.

Item	Group				
	Control	HNa	ZoA	SEM	*p*-Value
**CAT (U/mL)**	32.85 ^b^	39.31 ^c^	37.14 ^ab^	2.540	0.081
**T-AOC (U/mL)**	12.23 ^b^	14.99 ^c^	12.52 ^b^	0.950	0.033
**MDA (nmol/mL)**	5.71 ^b^	3.70 ^c^	4.32 ^a^	0.250	< 0.001

HNa—sodium humate; ZoA—zinc oxide + antibiotic; CAT—catalase; T-AOC—total antioxidant capacity; MDA—malondialdehyde. ^a, b, c,^ within a row, values with different letter superscripts mean significant difference (*p* < 0.05).

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
