# Peer review of "Effects of Dietary Supplementation of Humic Acid Sodium and Zinc Oxide on Growth Performance, Immune Status and Antioxidant Capacity of Weaned Piglets"

_animals, 2020, doi:10.3390/ani10112104_

Round 1
Reviewer 1 Report
Dear Editor in chief
The aim and objective of the article falls into the scope of the journal,
The article is well written, clear and deals with the fact [as authors mentioned in their manuscript] “the abuse of antibiotics and zinc oxide has caused problems such as antibiotic resistance and heavy metal residues”.
There is no novelty in the manuscript but can definitely contribute to the literature.
Some points that I would like to bring up prior to its acceptance:
L37: differences were not (you are talking about decreasing Conc. of both LPS and CAT)
L54: Although you have defined HNa previously in the abstract section, but please do again in the introduction section
L70: what was the grouping criteria? although Table 2 reveals that the treatment groups were iso-weighed but this point is also should be mentioned here.
L81: Table 1, Having feed formulation aboard, I would like to ask the authors to include the chemical composition (ME, CP, NDF, EE and EAA i.e. LYS, Met, etc) info into the table.
L82: The premix used in the basal diet (given to all three treatment groups) was already contained 100 mg of Zn as Zinc sulfate [other form of Zn inclusion rather than Zinc oxide], So Zn was received by all the animals in all treatment groups, which may cause unwanted interaction with the performances results.
For these kinds of studies, it’s important not to use commercial premixes. this point should be mention and discussed in discussion section.
L89: type of weight, model, country
L89: please ratify they feed consumption was measured for pen not individually as was done for weights
L93: by the same trained person? because these kinds of scoring are very subjective in nature, my score might be totally different than the score that is given by another person
L120: Table 2, As rule of thumb, "SEM" values should have an extra decimal respect to LSmeans values
L122: in case of producing parameters superscript "a" was given to smallest values where in F:G and Diarrhea rate it was given to highest values. please be consistent throughout the text
L128: Same comments as for table 2
L139: Same comments as for table 2 and 3
Discussion is well structured and concise, please mention and shortly discussed the use of Zinc sulfate in premix portion as I mentioned above.
Reviewer 2 Report
Wang et al. in their manuscript try to elucidate the effects of dietary supplementation of humic acid sodium and zinc oxide on growth performance, immune status and antioxidant capacity of weaned piglets. Although their work sounds interested the authors do not covered my espectations as they did not go further on the topic trying to combine the physiological parameters that they estimated with productive traits and why through i.e. a better antioxidative capacity a better performance is achieved. Thus, I reccoment a major revion of the manuscript trying to reach this approach, otherwise I feel that the manuscript lacks of novelty as the improvent of performance using HNa or ZnO is already well established.
In addition the manuscript needs further improvemnt in english style as well as in many linguistic points.
Another point that is very criticall is the final conclusion of the authors "It seems economically feasible to 226 add 2000 mg/kg HNa to the diet " it does not supported by any economical index throughout the manuscript either for the supplements or the livestock itself, thus it seems to be quite arbitrary.
Other points to be considered
line 15 : ommit and (affecting)
l. 17 Sodium Humate--> use small letters
l. 19 Anti-oxidative--> anti-oxidative
l. 21 Protect-->protect
l. 25 ommte parentheses
l. 28--> 3-->three
l. 33-40. Please rephrase to clearly state your conclusions
l. 61-->same with l. 19
l. 80--> 70%, respectively
l. 90-91--> does not make sense. use passive voice
l. 165--> ...due to the fact that...
l. 218: in vitro--> use italics
Round 2
Reviewer 2 Report
Authors improved their manuscript according to reviewrs' points both by terms of linguistic approach and of trying to combine their results with product traits and immuno characteristics. The fact that the specific mechanism of HNa is still unclear, and further research is needed highlights the importance of the presented results of the present study to this area. Thus, I recomend the manuscript entitled "“Effects of dietary supplementation of humic acid sodium and zinc oxide on growth performance, immune status and antioxidant capacity of weaned piglets” (animals-970300) for publication in animals journal.